# Symmetry-induced Disentanglement on Graphs

**Giangiacomo Mercatali**[*]
University of Manchester

**André Freitas**
Idiap Research Institute
University of Manchester

**Vikas Garg**
YaiYai Ltd and Aalto University
`vgarg@csail.mit.edu`

## Abstract

Learning disentangled representations is important for unraveling the underlying complex interactions between latent generative factors. Disentanglement has been formalized using a symmetry-centric notion for unstructured spaces, however, graphs have eluded a similarly rigorous treatment. We fill this gap with a new notion of *conditional symmetry* for disentanglement, and leverage tools from Lie algebras to encode graph properties into subgroups using suitable adaptations of generative models such as Variational Autoencoders. Unlike existing works on disentanglement, the proposed models segregate the latent space into uncoupled and entangled parts. Experiments on synthetic and real datasets suggest that these models can learn effective disengaged representations, and improve performance on downstream tasks such as few-shot classification and molecular generation.

## 1 Introduction

Disentanglement represents a fundamental desideratum in learning with limited supervision because it captures information about the salient (or explanatory) factors of variation in the data, and isolates information about each specific factor in only a few dimensions, thus unraveling the interactions underlying complex data [2, 59]. Disentangled representations have often been deemed responsible for providing neural models with the ability to improve performance on real-world downstream tasks [2, 20, 36, 58, 59]. Empirically, they have been shown to enable, or facilitate, critical properties such as sample efficiency [21] and generalization [32, 42, 69].

An interesting perspective on disentanglement has been recently proposed by Higgins et al. [20]. Specifically, they advocate viewing disentanglement as decomposition of the latent space of embeddings into subspaces, each of which can be transformed independently under the action of a single subgroup specific to the subspace, without affecting others. These subgroups arise, in turn, from the decomposition of a symmetry group. This formalism bestows several benefits, e.g., it (a) aligns with the idea that auxiliary information can be exploited for imposing a structure on the latent space [25, 40], and (b) leads to a rigorous theoretical framework for analyzing disentangled models [64, 70], e.g., those based on Variational Autoencoders (VAEs) [27, 50].

While symmetry groups provide a suitable tool both for formalizing [20] and learning [57, 64, 70] disentangled representations in unstructured domains, these techniques have not been investigated in the context of more general and complex data such as graphs. Furthermore, most works on disentanglement for graphs [1, 16, 31, 35, 38, 65] are not designed for generative settings, and the existing deep generative models (DGMs) for graph disentanglement such as [10, 17, 56] do not consider group symmetries. Note that differently from other definitions of disentanglement [2, 9, 11], leveraging group symmetries for disentanglement may unravel fundamental connections across several approaches based on graph neural networks (GNNs), i.e., the state-of-the-art models for embedding graphs [3, 5, 8, 13, 15, 39, 63].

---

[*]first-name.last-name@postgrad.manchester.ac.uk

36th Conference on Neural Information Processing Systems (NeurIPS 2022).

We, therefore, pursue two main goals in this work: (a) providing a rigorous symmetry-based formalism for disentanglement on graphs, and 2) designing novel neural architectures that are able to leverage symmetry groups to learn disengaged representations for graphs. However, accomplishing these objectives requires overcoming some challenges. In particular, graphs often abstract complex interactions, so the underlying latent space may not factorize completely into only disentangled subspaces. Thus, the symmetry-based formalism introduced in [20] does not suffice for graphs. Therefore, we introduce a new, more flexible notion of *conditional disentanglement*, advocating segregating the latent space into uncoupled *and* entangled parts. Translating this idea into an efficient algorithm requires further work, and we appeal to a Lie algebra based parameterization to encode the graph properties into subgroups.

**Contributions.** Our contributions can be summarized as follows:

- a novel notion of symmetry-induced conditional disentanglement (Section 3.1) that generalizes the previous definition from [20] (which we call unconditional disentanglement);

- a parameterization centered on Lie algebra based on the intuition that each separable generative factor can correspond to a Lie algebra coordinate element, and control a single graph property (Section 3.2);

- two algorithms for **S**ymmetry-**I**nduced **D**isentanglement under unconditional ($SID_U$) and conditional ($SID_C$) settings, both using a two-level variational auto-encoding mechanism, namely, one level for the data and other for symmetry group representation (Section 4); and

- a systematic evaluation on several disentanglement metrics [7, 11, 19, 26, 44]. The results demonstrate that the proposed models can outperform contemporary GNN baselines [38, 65]; successfully learn to decouple the entangled part of latent space from the disentangled parts (Section 5.1); and improve performance on downstream tasks such as few-shot classification, compression (Section 5.2) and molecular graph generation (Section 5.3).

We begin with a review of the relevant literature, and then proceed to the proposed framework.

## 2   Related work

**Symmetry-based disentanglement (SBD).** Initial works on SBD, such as Caselles-Dupré et al. [6], Quessard et al. [48], propose to use reinforcement learning to achieve irreducible representations of group elements through observation of action transitions. Painter et al. [44] introduce a VAE method that does not require labelled action-transition pairs, and demonstrate its validity within the environment from Caselles-Dupré et al. [6].

More recently, some algebraic approaches have been introduced. Zhu et al. [70] propose to decompose a Lie group into multiple subgroups, and enforce commutativity and independence between subgroups. Yang et al. [64] impose a cyclic group structure, encouraging the group to be isomorphic to the ground truth. Bouchacourt et al. [4] invoke group representation theory for defining distributed latent operators that are able to learn disentangled representations. Tonnaer et al. [57] provide a new metric for evaluation, and a VAE-based model for learning symmetry-based disentangled representations.

Some works leverage data manifolds for SBD. For example, Fumero et al. [12] view disentanglement as a product of low-dimensional sub-manifolds underlying the data space, such that each sub-manifold encodes an explainable data factor. Pfau et al. [46] propose a non-parametric algorithm that aims to discover a decomposition of the data manifold by investigating its holonomy group.

Other works explore SBD as well. For instance, Higgins et al. [22] demonstrate the importance of symmetry transformations and disentangled representation learning in the context of neuroscience. Wang et al. [61] design a contrastive self-supervised learning method that iteratively updates a partition, modeled as a symmetry group, until the considered factor is invariant.

**Disentangled GNNs.** To our knowledge, Ma et al. [38] provide the first graph disentanglement approach, which involves a routing mechanism aimed at separating the underlying factors. More recently, Yang et al. [65] exploit an attention mechanism for GNNs, and provide two evaluation metrics for disentanglement on graphs. Liu et al. [35] achieve disentanglement by encouraging

independence in the latent space, while Bae and Jeon [1] propose a method for disentangling multi-relational graphs motivated by an application on pedestrian trajectory prediction. Li et al. [31] provides a contrastive framework for disentangling the latent factors in GNNs.

Differently from these methods, we provide a formal definition of disentanglement for graphs, and leverage symmetry groups in a generative model setting.

**Graph DGMs.** Wu et al. [62], Yu et al. [68] propose VAE-based *Information Bottleneck* methods for improving the performance of GNNs on classification tasks, by learning minimal sufficient representations. The work by Simonovsky and Komodakis [55] aims to generate molecular graphs using graph VAEs. Liu et al. [34] adapt VAEs for conditioning molecule generation on specific properties. You et al. [67] propose a deep autoregressive model that learns to generate graphs by training on a representative set of graphs and decomposes the graph generation process into a sequence of node and edge formations. Du et al. [10] propose a DGM for spatio-temporal graphs, and Guo et al. [18] investigate DGMs with spatial networks. Finally, Guo et al. [17] and Stoehr et al. [56] investigate disentangled representations with graph VAEs.

## 3 Proposed framework

In Section 3.1 we review the unconditional symmetry-induced disentanglement, and propose a new definition of conditional disentanglement. In Section 3.2 we extend these notions to graphs. Finally, in Section 3.3, we describe additional loss terms to encourage disentanglement.

### 3.1 Symmetry-based disentanglement

**Definition 1** (Unconditional symmetry-based disentanglement [20])**.** Consider a generative process $b = W \rightarrow O$ mapping the *world states* $W$ into observations (i.e. data) $O$, and an inference process $h = O \rightarrow Z$ mapping the data into a latent vector space. By composing $b$ and $h$, we obtain $f = h \circ b$ such that $f : W \rightarrow Z$. Given group actions on $W$, i.e., $\cdot_W : G \times W \rightarrow W$ and on $Z$, i.e., $\cdot_Z : G \times Z \rightarrow Z$, the function $f$ is said to be *equivariant* between the actions on $W$ and $Z$ if the actions commute with $f$. We can express this equivariance mathematically as $g \cdot_Z f(w) = f(g \cdot_W w), \forall g \in G, \forall w \in W$. Assuming that the symmetry group $G$ can be decomposed into subgroups $G_1 \times \ldots \times G_n$, a vector representation $Z$ is disentangled with respect to this decomposition if all of the following conditions hold:

- there is an action defined on the representation set $\cdot_Z : G \times Z \rightarrow Z$,

- the map $f : W \rightarrow Z$ is equivariant between the actions on $W$ and $Z$, and

- there is a decomposition $Z = Z_1 \times Z_2 \ldots \times Z_n$ such that each $Z_i$ is affected only by $G_i$ and is invariant to (i.e., does not change due to) $G_j$ for all $j \neq i$.

Definition 1 does not allow any part of the latent vector space to be entangled, which is likely to be restrictive for complex applications. Therefore, we propose conditional disentanglement.

**Definition 2** (Conditional symmetry-based equivariance)**.** Consider a generative process $b = W \times E \rightarrow O$ mapping the disentangle-able world states $W$ and non-disentangled states $E$ into observations (i.e. data) $O$, and an inference process $h = O \rightarrow Z$ mapping the data into a latent vector space. By composing $b$ and $h$, we obtain $f = b \circ h$, which maps $W \times E$ to $Z$. Given group actions on $W$, namely, $\cdot_W : G \times W \rightarrow W$ and on $Z$, namely, $\cdot_Z : G \times Z \rightarrow Z$, the function $f$ is said to be *conditionally equivariant* between the actions on $W$ and $Z$ given $E$ if the actions commute with $f$. That is, $g \cdot_Z f(w, e) = f(g \cdot_W w, e), \forall g \in G, \forall w \in W, \forall e \in E$.

This definition of conditional equivariance naturally leads to a new notion of *conditional disentanglement* using a characterization similar to the three conditions for the unconditional case. Specifically, we need to impose the condition that $f$ is conditionally equivariant between the actions of $W$ and $Z$ given $E$ (instead of simply being equivariant between the actions of $W$ and $Z$).

Note that while this definition results in a generalized framework for disentanglement, it does not directly translate into a learning algorithm. Toward that goal, we invoke tools from Lie algebra as described in the next section.

## 3.2 Disentanglement on graphs

Here, we aim to instantiate Definition 1 and 2 in the context of models that encode graph-structured data. Drawing inspiration from the previous work on Lie groups and algebras for symmetry-induced disentangled VAEs by Zhu et al. [70], we develop a formalism for disentanglement on graphs induced by Lie groups.

**Lie groups and Lie algebras.** A Lie group $G$ is a group of continuous symmetries [52], and associated with a Lie algebra $\mathfrak{g}$, which is the tangent space to the identity element of $G$. We can thus parameterize a Lie algebra with a basis $\{\mathbb{A}_i\}_{i=1}^k$, where every element in $\mathfrak{g}$ can be written as $\mathbb{A} = \mathbb{A}_1 t_1 + \ldots + \mathbb{A}_k t_k$ using coordinates $t_i$. Elements of the Lie algebra can be mapped back into the Lie group with a matrix exponential map $\exp : \mathfrak{g} \to G$.

We now provide some intuition into how we can connect the notion of a Lie group $G$ with the definition of disentanglement in a graph network, by associating each Lie algebra coordinate $t_i$ with a generative factor. A latent representation $\hat{Z}$, obtained, e.g., from a graph encoder, is disentangled with respect to a Lie group $G$, if a change in the coordinate $t_i$ is associated with a change in only the $i^{th}$ component of $\hat{Z}$, i.e., only $\hat{z}_i$. In other words, disentanglement entails that the semantics $(t_1, t_2, \ldots, t_k)$ are equivariant with respect to the properties $(\hat{z}_1, \hat{z}_2, \ldots, \hat{z}_k)$ that are being predicted.

Next, we provide the formal definitions for unconditional and conditional graph disentanglement. The former expresses $\hat{Z}$ as a function of $T$ without accounting for $Z$, whereas the latter expresses $\hat{Z}$ based on $T$ after fixing an encoding $Z$.

**Definition 3** (Lie-algebra based unconditional graph disentanglement). The graph embedding $\hat{Z}$ obtained by $f(\hat{Z}|T)$ is *unconditionally disentangled* with respect to the Lie group coordinates $T = \{t_j\}_{j=1}^k$ if the following hold: (a) there is a group action $\cdot_{\hat{Z}} : G \times \hat{Z} \to \hat{Z}$ on $\hat{Z}$, (b) the map $f = \exp(\mathbb{A}(T)) : T \to \hat{Z}$ is equivariant between actions on $T$ and $\hat{Z}$, and (c) there is a decomposition $\hat{Z} = \hat{z}_1 \times \hat{z}_2 \ldots \times \hat{z}_k$, where each coordinate $t_i$ affects only the corresponding component $\hat{z}_i$.

**Definition 4** (Lie-algebra based conditional graph disentanglement). The graph embedding $\hat{Z}$ obtained by $f(\hat{Z}|T, Z)$ is *conditionally disentangled* with respect to the Lie group coordinates $T = \{t_j\}_{j=1}^k$ given $Z$ if (a) there is a group action $\cdot_{\hat{Z}} : G \times \hat{Z} \to \hat{Z}$ on $\hat{Z}$, (b) the map $f : \exp \mathbb{A}(T) \times Z \to \hat{Z}$ is equivariant between actions on $T$ and $\hat{Z}$ for any fixed $Z$, and (c) $\exp \mathbb{A}(T) \times Z$ factorizes into the product $\prod_{i=1}^k \exp(t_i \mathbb{A}_i) \times Z$ such that each component $\exp(t_i \mathbb{A}_i)$ is affected only by the corresponding coordinate $t_i$ for $i \in \{1, 2, \ldots, k\}$ for any fixed $Z$.

## 3.3 Disentanglement constraints

It is known [64, 70] that a symmetry group parameterization is usually not sufficient for a group to be decomposed into subgroups that are parameterized independently by a single coordinate. Therefore, following Zhu et al. [70], we add two additional constraints for enforcing disentanglement. These constraints can be relaxed, and included in the loss function as regularization terms.

**Commutative penalty.** We would like to enforce a coordinate $t_i$ to be identified by single group $\exp(t_i \mathbb{A}_i)$, and thus represent a single property variation in the data $\hat{z}_i$. Mathematically, one can state this desideratum as an equivalence between the exponential map of the sum, and the product over exponential maps. [70] proved this to be achieved under commutativity over the Lie algebra basis. Namely, if $\mathbb{A}_i \mathbb{A}_j = \mathbb{A}_j \mathbb{A}_i$ then $\exp\left(\sum_{i=1}^k t_i \mathbb{A}_i\right) = \prod_{i=1}^k \exp(t_i \mathbb{A}_i)$.

**Hessian penalty** Furthermore, disentanglement can be encouraged using a Hessian penalty [45] based on the fact the Hessian matrix with respect to a disentangled representation is always zero (owing to the independence between different dimensions). Zhu et al. [70] adapt this penalty in a Lie algebra parameterization setup, as $H_{ij} = \frac{\delta^2 g(T)}{\delta t_i \delta t_j}$ where $g(T) = \exp\left(\sum_{i=1}^k t_i \mathbb{A}_i\right)$, and show that if $\mathbb{A}_i \mathbb{A}_j = 0 \ \forall i, j$ then $H_{ij} = 0$.

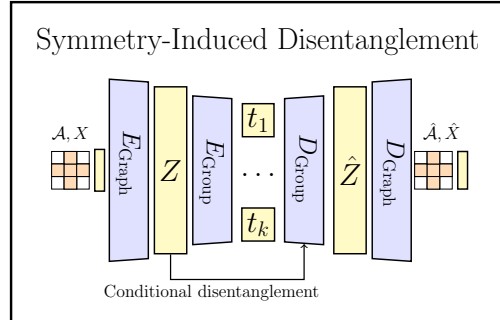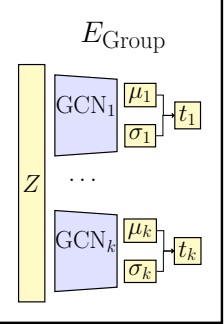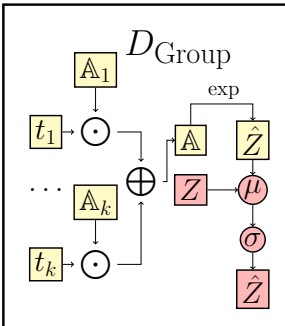

Figure 1: **Left** (architecture): SID is composed of 4 blocks, i.e., 2 layers each for the auto-encoding modules for graph data and group structure. **Middle** ($E_{\text{group}}$): the Lie algebra coordinates $t_i$ are obtained by applying $k$ graph convolutional networks (GCNs) to the graph embedding $Z$. **Right** ($D_{\text{group}}$): the coordinates $t_i$ are combined with basis elements $\mathbb{A}$ to reconstruct the graph latent code. In the conditional case (red nodes) we further condition on $Z$, by taking the mean ($\mu$) of $Z$ and $\hat{Z}$, and passing $\mu$ through an activation function $\sigma$ (ReLu for our experiments).

## 4 Method

We propose Symmetry-Induced Disentanglement (SID), a VAE framework for graphs [28, 55] which encourages disentangled representations using a Lie group parameterization and two regularization losses (commutative and hessian penalties). The architecture is depicted in Figure 1.

### 4.1 Probabilistic formulation

We use the following notation for model variables: latent graph embeddings $Z$ (and $\hat{Z}$), graph features $X$, symmetry Lie group structure $T$, and the graph adjacency matrix $\mathcal{A}$. We next derive lower bounds on log-likelihood for both the unconditional and conditional settings (see Appendix for details).

**Proposition 1** (Unconditional lower bound). Given two latent variables $Z$ and $T$, we instantiate the unconditional disentanglement from Definition 3 with a probabilistic model that maximizes the log-likelihood of the graph data $\mathcal{G} = (X, \mathcal{A})$ by optimizing the following lower bound:

$$\mathcal{L}_{\text{u}} = \mathbb{E}_{q(Z|\mathcal{G})q(T|Z,\mathcal{A})} \log p(Z|T)p(\mathcal{G}|Z) - \mathbb{E}_{q(Z|\mathcal{G})} \text{KL}(q(T|Z,\mathcal{A})||p(T)) - \mathbb{E}_{q(Z|\mathcal{G})} \log q(Z|\mathcal{G}) .$$
(1)

**Proposition 2** (Conditional lower bound). We extend Proposition 1 to account for conditional disentanglement as defined in Definition 4. We obtain the following lower bound

$$\mathcal{L}_{\text{c}} = \mathbb{E}_{q(Z|\mathcal{G})q(T|Z,\mathcal{A})} \log p(\hat{Z}|T,Z)p(\mathcal{G}|\hat{Z}) - \mathbb{E}_{q(Z|\mathcal{G})} \text{KL}(q(T|Z,\mathcal{A})||p(Z,T)) - \mathbb{E}_{q(Z|\mathcal{G})} \log q(Z|\mathcal{G}) .$$
(2)

**Architecture.** Propositions 1 and (2) are derived respectively in Appendix. The respective losses (1 and 2) are implemented with four neural network modules as follows. Two stochastic encoders ($E_{\text{graph}}$ and $E_{\text{group}}$) embed respectively the graph data $\mathcal{G}$ into a latent variable $Z$ via $q(Z|\mathcal{G})$, and $Z$ into the Lie group coordinates $T$ via $q(T|Z)$. Two deterministic decoders ($D_{\text{group}}$ and $D_{\text{graph}}$) reconstruct, respectively, the graph embedding $\hat{Z}$ from $T$ via $p(\hat{Z}|T)$ for the unconditional case, and the graph $\mathcal{G}$ from $\hat{Z}$ via $p(\mathcal{G}|\hat{Z})$. The decoder network $D_{\text{group}}$ also plays a role in implementing the conditional lower bound, via $p(\hat{Z}|Z,T)$, which computes $\hat{Z}$ conditioned on $Z$ and $T$.

### 4.2 Encoders

**Graph encoder.** We first encode the graph features $X$ and the adjacency matrix $\mathcal{A}$ in the latent space $Z$, using an inference network $E_{\text{graph}}$ $q(Z|X,\mathcal{A})$. Similarly to Kipf and Welling [28], we use a single 2-layer network, where the first layer leverages a graph convolutional network (GCN) to reduce the dimension of the graph features to the symmetry group size: $\overline{X} = \text{GCN}(X,\mathcal{A})$. The second layer computes the mean and variance vectors ($\mu = \text{GCN}_\mu(\overline{X},\mathcal{A})$, $\log \sigma = \text{GCN}_\sigma(\overline{X},\mathcal{A})$), which are

then used to sample the latent variable $Z$ from a Gaussian distribution with the *reparameterization trick*: $q(Z|X, \mathcal{A}) = \mathcal{N}(Z|\mu, \text{diag}(\sigma^2))$.

**Group encoder.** We employ $E_{\text{group}}$ to map the latent vector $Z$ into a vector $T$ of $k$ Lie algebra coordinates, where $k$ is the subspace size (e.g., the number of disentangled elements). In our settings, each lie algebra coordinate $\{t_i\}_{i=1}^k$ is the output of a GCN with two layers. The first one computes a graph embedding $\overline{Z} = \text{GCN}(Z, \mathcal{A})$, and the second one produces the vectors $\mu = \text{GCN}_\mu(\overline{Z}, \mathcal{A})$, $\log \sigma = \text{GCN}_\sigma(\overline{Z}, \mathcal{A})$, which in turn yield produce $\{t_i\}_{i=1}^k$ with the reparameterization trick. Finally, $\{t_i\}_{i=1}^k$ are combined into a single tensor $T$ using a multilayer perceptron (MLP):

$$q(T|Z, \mathcal{A}) = \text{MLP}(q(t_i|Z, \mathcal{A})) \quad \text{with} \quad q(t_i|Z, \mathcal{A}) = \mathcal{N}(t_i|\mu_i, \text{diag}(\sigma_i^2)) \text{ for } i = 1 \ldots, k \quad (3)$$

### 4.3 Decoders

**Group decoder.** The module $D_{\text{group}}$ reconstructs the latent variable $\hat{Z}$ using a Lie group $G$ and a Lie algebra $\mathfrak{g}$. Specifically, we first learn a Lie algebra basis element $\{\mathbb{A}_i\}_{i=1}^k \in \mathfrak{g}$ for each coordinate $t_i$. We then aggregate the coordinates $t_i$ and basis elements $\mathbb{A}_i$ with an exponential map, to obtain a group representation as

$$p(\hat{Z}|T) = g(T) = \exp(\mathbb{A}(T)) \quad \text{where} \quad \mathbb{A}(T) = \sum_{i=1}^k t_i \mathbb{A}_i \quad \text{for } g \in G, \ \mathbb{A} \in \mathfrak{g} \, . \quad (4)$$

The conditional version of $D_{\text{group}}$ is obtained with a slight modification of Eq. 4: $\hat{Z}$ is computed by feeding the mean between $Z$ (e.g. the graph embeddings obtained from $E_{\text{graph}}$), and $\exp(\mathbb{A}(T))$ into a non-linear activation function $\sigma$ such as ReLU:

$$p(\hat{Z}|T, Z) = \sigma(\text{mean}[\exp(\mathbb{A}(T)), Z]), \quad \text{where} \quad Z = \{z_j\}_{j=1}^k \, . \quad (5)$$

**Graph decoder.** The network $D_{\text{graph}}$ reconstructs the graph data $\hat{\mathcal{G}}$ from the latents $\hat{Z}$. This is achieved by first passing the latent feature to a GCN layer that maps $Z$ into a vector of the same dimension as the original feature vectors, and subsequently into a sigmoid activation function:

$$p(\mathcal{A}, X|\hat{Z}) = p(\mathcal{A}, X|\sigma(\text{GCN}(\hat{Z}))) \quad (6)$$

### 4.4 Training

The proposed unconditional and conditional models are trained by optimizing the loss given by $\beta \mathcal{L} + \lambda h + \gamma c$, where $\mathcal{L}$ is the lower bound from Eq. 1 (unconditional) or Eq. 2 (conditional), and $h$ and $c$ are respectively the hessian and commutative penalties, obtained as regularization terms adapting the implementation from Zhu et al. [70]. We provide insights into the effect of parameters on the models and their performance with an ablation study in Section 5.1. For neural architecture, we set each encoder and decoder to consist of 3 layers, where each layer takes as input the graph features from the previous layer.

## 5 Experiments

We conducted extensive experiments that we decribe now. Section 5.1 evaluates the disentanglement capabilities of the models, Section 5.2 provides a compression and few-shots classification experiment, and Section 5.3 assesses the generation capabilities on molecular datasets.

### 5.1 Disentanglement evaluation

We evaluate disentanglement with a) an ablation study for assessing our models components, b) qualitative investigations, and c) quantitative metrics. For metrics, we follow the evaluation protocols from Locatello et al. [36], which involve being able to randomly combine a number of generative factors to sample a new data. In our case, we provide the necessary generative factors to generate two types of synthetic random graphs (e.g. Erdos-Renyi and Watts-Strogatz). More details about our dataset construction procedure are provided in Appendix A. We compute 5 metrics, including $\beta$-VAE [19] (Beta), FactorVAE (FVM) [26], Mutual Information Gap (MIG) [7], DCI Disentanglement [11], and Factor Leakage (FL) [44].

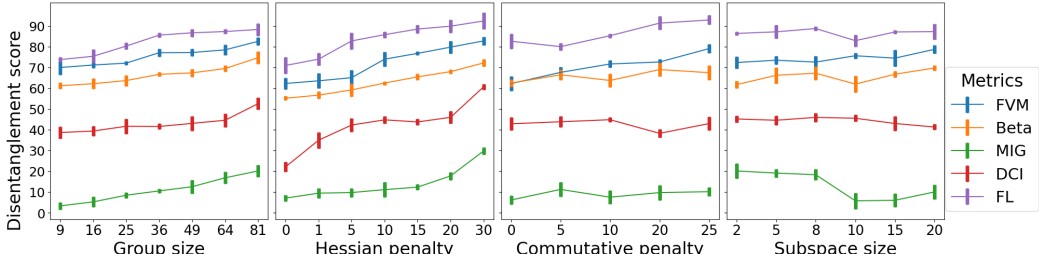

Figure 2: Impact of single model components on disentanglement metrics.

**Ablation study.** We start by evaluating the impact of our model components on the disentanglement metrics, by maintaining all parameters fixed, except from one that varies. We probe the effect from different group sizes, hessian penalties, commutative penalties, and subspace sizes. In Figure 2 we report the mean over 10 runs for our unconditional model on the synthetic Watts-Strogatz dataset. We observe that when the group size is increased, there is an improvement in disentanglement across all metrics, and this can be explained by the idea that with a larger group size, the subgroups, represented by the Lie algebra coordinate $t_i$ can control different data variations. The hessian penalty results to be effectively enhancing disentangled representations, while the commutative penalty is less effective, and we can explain this result because the former requires the Lie algebra basis elements to have mutual products of zeros while subgroup decomposition only requires their commutators to be zeros, which also confirms the results obtained for image datasets [70].

**Correlation analysis.** Following previous work [65, 38], we report the correlation matrix of the hidden graph features, the goal of which is to show whether the features capture mutually exclusive information, which is achieved when the there is a block-wise correlation pattern. In our models we use a group-size of 32 (to match with hidden features size of the other methods), a hessian penalty of 40 and commutative penalty of 5. In Figure 3 we note that both our $SID_U$ and $SID_C$ models achieve a block-wise correlation pattern, however the conditional version has also some other highlighted regions, because in this model involves fixing a latent vector $Z$ to condition the reconstruction. As a result, the conditioning may be responsible for the visualized correlation on the hidden features.

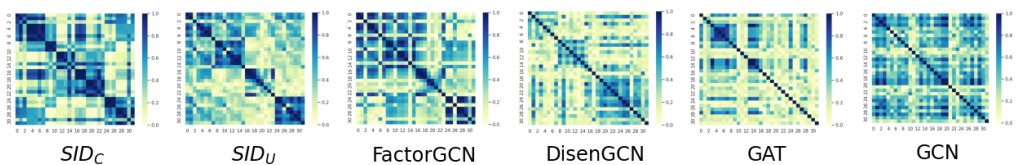

$SID_C$      $SID_U$      FactorGCN      DisenGCN      GAT      GCN

Figure 3: Correlation analysis

**Learned graph variations.** In this experiment, we train the $SID_C$ model on Watts-Strogatz (WS) random graphs, which have 3 generative factors, $n$ for number of nodes, $p$ for connectivity (the probability of rewiring each edge), and $k$ for neighborhood ($k$ nearest neighbors in a ring topology). We adapt our model to the visualization implementations provided by Stoehr et al. [56], which report adjacency matrices (black) and graph (white) information, where variations in node attribute values are indicated in blue. In Figure 4 we report the results for the $SID_C$ model by projecting pairs of latent variable traversals to see how the factors vary, and we observe the following. 1) The variable $z_0$ controls the node attribute value. For example, in `adj`$_{01}$ we see the color changing from blue to white along the horizontal axis. 2) The variable $z_1$ controls connectivity, for example in `graph`$_{01}$, in the $z_1$ axis (top to bottom) we see that the number of edges decrease, while in `graph`$_{12}$, along the $z_1$ axis (left to right), the number of edges increase. 3) The variable $z_2$ controls the number of nodes, and we see that in `graph`$_{12}$ and `graph`$_{02}$ the number of nodes decreases along the $z_2$ axis (bottom to top).

**Disentanglement metrics.** We perform 10 random seed runs, by training for 200 epochs on the proposed synthetic datasets and measure the quantitative metrics on 1000 data points. In terms of

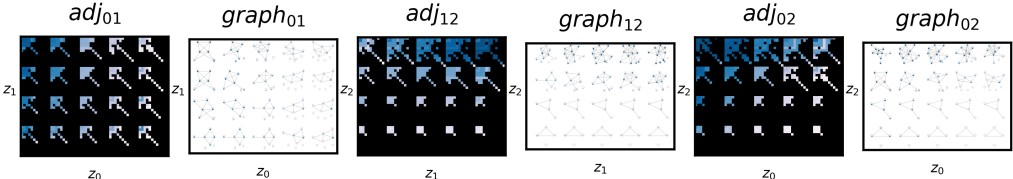

Figure 4: Graph traversals (attributes and node values) for the combinations of 3 latent variables.

Table 1: Disentanglement results.

| Model | Watts-Strogatz | | | | | Erdos-Renyi | | | | |
|---|---|---|---|---|---|---|---|---|---|---|
| | FVM | Beta | MIG | DCI | FL | FVM | Beta | MIG | DCI | FL |
| GCN | $69.6_{\pm0.8}$ | $66.5_{\pm0.7}$ | $23.2_{\pm2.5}$ | $43.8_{\pm1.6}$ | $84.1_{\pm0.5}$ | $57.0_{\pm4.5}$ | $61.0_{\pm1.4}$ | $22.5_{\pm3.3}$ | $54.1_{\pm2.1}$ | $71.7_{\pm2.6}$ |
| GAT | $72.6_{\pm1.3}$ | $66.2_{\pm3.4}$ | $17.4_{\pm1.0}$ | $53.2_{\pm2.7}$ | $79.0_{\pm1.1}$ | $56.0_{\pm3.2}$ | $66.3_{\pm2.5}$ | $31.4_{\pm1.9}$ | $54.0_{\pm2.9}$ | $70.2_{\pm2.1}$ |
| FAC | $74.2_{\pm1.3}$ | $65.7_{\pm2.6}$ | $26.4_{\pm2.0}$ | $53.4_{\pm4.0}$ | $80.3_{\pm0.9}$ | $67.2_{\pm1.7}$ | $66.3_{\pm3.3}$ | $35.1_{\pm2.4}$ | $53.8_{\pm1.5}$ | $78.1_{\pm3.0}$ |
| DIS | $73.0_{\pm3.5}$ | $69.3_{\pm2.1}$ | $27.1_{\pm1.7}$ | $52.2_{\pm3.9}$ | $85.2_{\pm2.0}$ | $68.1_{\pm1.8}$ | $64.1_{\pm0.5}$ | $41.3_{\pm1.4}$ | $53.5_{\pm2.3}$ | $72.4_{\pm2.3}$ |
| $SID_U$ | $80.5_{\pm1.5}$ | $71.6_{\pm4.5}$ | $28.5_{\pm2.8}$ | $55.4_{\pm2.0}$ | $91.0_{\pm2.3}$ | $83.0_{\pm2.9}$ | $71.8_{\pm3.2}$ | $\mathbf{53.2}_{\pm0.8}$ | $58.6_{\pm1.9}$ | $87.1_{\pm4.0}$ |
| $SID_C$ | $\mathbf{81.9}_{\pm1.2}$ | $\mathbf{74.5}_{\pm3.2}$ | $\mathbf{49.1}_{\pm3.1}$ | $\mathbf{59.7}_{\pm1.2}$ | $\mathbf{92.3}_{\pm0.9}$ | $\mathbf{83.3}_{\pm0.7}$ | $\mathbf{74.1}_{\pm1.2}$ | $52.4_{\pm3.1}$ | $\mathbf{60.5}_{\pm1.2}$ | $\mathbf{89.8}_{\pm2.3}$ |

baselines, we consider graph VAEs with GAT [60] and GCN [29] encoders, as well as disentangled encoders such as FAC [65] and DIS [38]. We set our models to have a group size of 81, a hessian penalty of 40, and a commutative penalty of 5.

In Table 1 we observe that our models are able to improve on previous baselines on all metrics on the evaluated datasets, and the conditional version achieves the best performance in most cases. We notice that the FL metric shows a significant performance increase, justified by its ability to capture the disentanglement on each generative factor, which, in our models is controlled by the group size, while in previous approaches is not considered. In terms of the other previously proposed disentangled approaches, DIS and FAC demonstrate higher disentanglement in most cases, when compared to GCN and GAT, however they are outperformed by our methods.

## 5.2 Compression and few-shots classification

Following the setup from Ge et al. [14], we consider the task of learning a compressed model via multiple pooling operations, and test on few-shots classification capabilities on datasets with multiple classes, including FRANKENSTEIN [43] (2 classes), COLORS-3 [30] (11 classes), Mutagenicity [51] (2 classes), NCI1 [53] (2 c). The baselines include MIA [14], which propose an attention-based pooling for compression, as well as GCN [29], and GAT [60] based autoencoders, equipped with the same pooling layer. All models are set to have a pooling rate of 0.8, and a depth of 3 layers. Our models are set with a group size of 81, hessian penalty of 20 and commutative penalty of 5. The task consists in training first without labels to learn the representation, and then using the compressed model as input for training a GCN classifier a labelled subset of the dataset, which involves 100 samples for training and 100 for testing.

In Table 2 we report the mean and variance across 10 runs for MSE, the classification accuracy, and the size of the trained model. The results show that $SID_C$ and $SID_U$ outperform the considered baselines in terms of few-shots classification accuracy on all datasets, while achieving a smaller compressed size. We hypothesize that the superior accuracy performance of our models is due to a more efficient learning provided by disentangled representations, which have previously demonstrated the ability to enhance the accuracy of predictions in the context of generalization tasks [41, 69, 32]. In terms of MSE, our models show an improvement compared to GCN- and GAT-based autoencoders, but they are outperformed by MIA in most cases.

Table 2: Compression and few shot classification results.

| Model | FRANKENSTEIN | | | COLORS-3 | | | Mutagenicity | | | NCI1 | | |
|---|---|---|---|---|---|---|---|---|---|---|---|---|
| | MSE | Size | Acc. | MSE | Size | Acc. | MSE | Size | Acc. | MSE | Size | Acc. |
| MIA | $1.6_{\pm0.5}$ | 4.9M | $65_{\pm1}$ | $1.8_{\pm0.2}$ | 729K | $33_{\pm2}$ | $4.1_{\pm0.8}$ | 753K | $78_{\pm2}$ | $3.2_{\pm0.5}$ | 777K | $61_{\pm1}$ |
| GCN | $12.6_{\pm2.5}$ | 4.2M | $39_{\pm3}$ | $11.2_{\pm1.1}$ | 657K | $33_{\pm2}$ | $16.5_{\pm2.1}$ | 673K | $72_{\pm1}$ | $7.4_{\pm1.2}$ | 697K | $53_{\pm4}$ |
| GAT | $9.0_{\pm1.2}$ | 4.2M | $26_{\pm2}$ | $10.1_{\pm3.1}$ | 665K | $30_{\pm1}$ | $11.3_{\pm2.8}$ | 681K | $79_{\pm4}$ | $8.1_{\pm0.3}$ | 705K | $50_{\pm2}$ |
| $SID_U$ | $4.3_{\pm1.1}$ | **4.1M** | $68_{\pm2}$ | $5.1_{\pm1.5}$ | **625K** | $39_{\pm1}$ | $4.2_{\pm0.3}$ | 633K | $81_{\pm3}$ | $3.5_{\pm0.3}$ | **657K** | $63_{\pm2}$ |
| $SID_C$ | $2.8_{\pm0.5}$ | **4.1M** | $69_{\pm2}$ | $1.9_{\pm0.4}$ | **625K** | $38_{\pm3}$ | $3.2_{\pm1.1}$ | 633K | $84_{\pm2}$ | $3.9_{\pm0.9}$ | **657K** | $66_{\pm2}$ |

Table 3: Random graph generation.

| Model | ZINC | | | | QM9 | | | | MOSES | | | |
|---|---|---|---|---|---|---|---|---|---|---|---|---|
| | Val | WR | Uni | Nov | Val | WR | Uni | Nov | Val | WR | Uni | Nov |
| JT-VAE | **100** | n/a | **100** | **100** | n/a | n/a | n/a | n/a | **100** | n/a | **99.96** | 91.43 |
| GCPN | **100** | 20 | 99.97 | **100** | n/a | n/a | n/a | n/a | n/a | n/a | n/a | n/a |
| GraphAF | **100** | 68 | 99.1 | **100** | **100** | 67 | 94.15 | 88.83 | **100** | 71 | 99.99 | **100** |
| GraphDF | **100** | **89.03** | 99.16 | **100** | **100** | 82.67 | **97.62** | **98.1** | **100** | **87.58** | 99.55 | **100** |
| $SID_U$ | **100** | 78.21 | 99.12 | **100** | **100** | 75.43 | 95.22 | 90.33 | **100** | 81.62 | 99.62 | **100** |
| $SID_C$ | **100** | 86.02 | 99.14 | **100** | **100** | **82.92** | 96.43 | 92.28 | **100** | 84.29 | 99.79 | **100** |

## 5.3 Molecular graph generation

Molecular generation involves three tasks, random generation, property optimization and constrained optimization, and a suitable setup can be obtained following the approaches implemented via Torch-Drug[2] and DIG [33]. The generation process involves 1) learning the distribution of the molecular data and 2) fine-tuning the pre-trained model on one of the tasks. We employ our models during the training phase, using a group size of 81, a hessian penalty of 40 and commutative penalty of 5, and then we follow the reinforcement learning fine-tuning method from GCPN [66]. Similar setups are used from previous molecule generation methods, which we include as baselines: JT-VAE [24], GraphAF [54], GraphDF [37], GCPN [66].

**Random generation.** This task evaluates the quality of randomly generated samples on 4 standard metrics in percentage, including: valid molecules with resampling (Val), valid molecules without resampling (WR), unique molecules (Uni), novel molecules that not appear in the training data (Nov). We train our models for 10 epochs, with a batch size of 32 and the learning rate of 0.001, and compute the metrics over 10K generated molecules. In Table 3 we report results on ZINC [23], QM9 [49], and MOSES [47]. We observe that both $SID_U$ and $SID_C$ are able to improve the performance of GraphAF and GCPN on most metrics, while achieving comparable performance with GraphDF, furthermore the $SID_C$ model achieves the highest score of validity without resampling on the QM9 dataset. Since our model is not optimized for molecule generation, the positive results of our models indicate that disentangled representations may represent a flexible tool for generating more realistic graphs, and that encouraging the separation of semantic factors into different latent codes, may be beneficial for improving the quality of the samples.

**Property optimization** The goal of the task is to generate novel molecules with high property scores for the penalized logP property, and the quantitative estimation of drug-likeness property (QED). The setup involves pretraining our models for 300 epochs on ZINC random generation, applying the RL fine-tuning procedure from GCPN, and measuring the scores from the top 3 generated molecules. In Table 4 we observe that our models are able to outperform GCPN and GraphAF for both penalized logP and QED, and achieve comparable performance to the more advanced GraphDF model. In Figure 5 we show the molecules generated using $SID_C$. While our model does not outperform GraphDF, our results are relevant because they demonstrate that the disentangled representations learned from our models can be leveraged to improve the performance of the downstream task of property optimization.

---

[2]https://torchdrug.ai/ (Apache license)

Table 4: Property optimization performance.

| | Penalized logP | | | QED | | |
|---|---|---|---|---|---|---|
| | 1st | 2nd | 3rd | 1st | 2nd | 3rd |
| ZINC | 4.52 | 4.3 | 4.23 | 0.948 | 0.948 | 0.948 |
| JT-VAE | 5.3 | 4.93 | 4.49 | 0.925 | 0.911 | 0.910 |
| GCPN | 7.98 | 7.85 | 7.80 | **0.948** | 0.947 | 0.946 |
| GraphAF | 12.23 | 11.29 | 11.05 | **0.948** | **0.948** | 0.947 |
| GraphDF | **13.7** | **13.18** | **13.17** | **0.948** | **0.948** | **0.948** |
| $SID_U$ | 12.56 | 12.46 | 12.03 | 0.947 | **0.948** | 0.947 |
| $SID_C$ | 12.89 | 12.82 | 12.27 | **0.948** | **0.948** | **0.948** |

Figure 5: Molecules from $SID_C$

Table 5: Constrained optimization performance on 800 molecules used in GraphAF.

| $\delta$ | GraphAF | | | GraphDF | | | $SID_C$ | | |
|---|---|---|---|---|---|---|---|---|---|
| | Imp | Sim | Suc | Imp | Sim | Suc | Imp | Sim | Suc |
| 0.0 | $13.13_{\pm 6.89}$ | $0.29_{\pm 0.15}$ | 100 | $\mathbf{14.15}_{\pm 6.86}$ | $0.29_{\pm 0.13}$ | 100 | $13.26_{\pm 3.24}$ | $0.28_{\pm 0.21}$ | 100 |
| 0.2 | $11.90_{\pm 6.86}$ | $0.33_{\pm 0.12}$ | 100 | $\mathbf{12.77}_{\pm 6.59}$ | $0.32_{\pm 0.11}$ | 100 | $12.35_{\pm 5.62}$ | $0.32_{\pm 0.13}$ | 100 |
| 0.4 | $8.21_{\pm 6.51}$ | $0.49_{\pm 0.09}$ | 99.8 | $\mathbf{9.19}_{\pm 6.43}$ | $0.48_{\pm 0.08}$ | 99.6 | $8.25_{\pm 4.93}$ | $0.50_{\pm 0.18}$ | 98.4 |
| 0.6 | $\mathbf{4.98}_{\pm 6.49}$ | $0.66_{\pm 0.05}$ | 96.8 | $4.51_{\pm 5.80}$ | $0.65_{\pm 0.05}$ | 92.1 | $4.67_{\pm 5.39}$ | $0.66_{\pm 0.03}$ | 93.2 |

**Constrained Optimization.** This task aims to modify the input molecular graph for improving its penalized logP score while keeping the similarity between the input and modified molecules higher than the threshold $\delta$. We follow the setup of GraphAF and GraphDF, which select 800 molecules from ZINC with low penalized logP scores as the input molecules to be optimized. We pretrain our model on the random generation task for 300 epochs, and then fine-tune with the RL procedure from GraphAF. We report the mean and standard deviation of the largest property improvement (Imp), and similarities (Sim) between them and their corresponding input molecules, as well as the success rate (Suc). In Table 5 we compare the results for our $SID_C$ model with GraphAF and GraphDF. We observe that $SID_C$ is able to outperform GraphAF 3 out of 4 times for improvement values, and achieves comparable results with GraphDF, which we motivate by the fact that GraphDF provides further fine-tuning in order to improve the baseline. In terms of similarity and success rate, our model has comparable results with the baselines for all $\delta$ setups.

## 6 Discussion

**Limitations.** For quantitative evaluation, we rely on synthetic data, since current metrics are based on the ability to sample the factors and combine them into observations Kim and Mnih [26]. Future work should investigate how to provide metrics that can evaluate disentanglement on real-world datasets. A recent line of work from Khemakhem et al. [25] shows that disentanglement is closely related to model identifiability, i.e., a representation is disentangled when a set of learned parameters is uniquely identified with the parameters in the true distribution. Our approach has not considered or provided proofs for, identifiability, and we leave this as future work.

**Conclusion.** We formalize the notion of conditional disentanglement on graphs and propose a novel framework for graph disentanglement by leveraging tools from Lie algebras. Based on the new definition of disentanglement, we design a graph VAE based on a Lie group parameterization, and provide a novel ELBO criteria for optimizing conditional disentanglement. Our method achieves superior performance on quantitative disentanglement benchmarks when compared to contemporary disentangled GNNs and other convolutional layers. Finally, we demonstrate strong capabilities on few-shots classification and molecular generation experiments.

## 7 Acknowledgements

We thank the anonymous reviewers for their helpful comments. GM's research was partially funded by EPSRC and the BBC under iCASE.

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
