| $\mathbf{1.6}_{\pm 0.5}$ | 4.9M | $65_{\pm 1}$ | $\mathbf{1.8}_{\pm 0.2}$ | 729K | $33_{\pm 2}$ | $4.1_{\pm 0.8}$ | 753K | $78_{\pm 2}$ | $\mathbf{3.2}_{\pm 0.5}$ | 777K | $61_{\pm 1}$ |
| GCN | $12.6_{\pm 2.5}$ | 4.2M | $39_{\pm 3}$ | $11.2_{\pm 1.1}$ | 657K | $33_{\pm 2}$ | $16.5_{\pm 2.1}$ | 673K | $72_{\pm 1}$ | $7.4_{\pm 1.2}$ | 697K | $53_{\pm 4}$ |
| GAT | $9.0_{\pm 1.2}$ | 4.2M | $26_{\pm 2}$ | $10.1_{\pm 3.1}$ | 665K | $30_{\pm 1}$ | $11.3_{\pm 2.8}$ | 681K | $79_{\pm 4}$ | $8.1_{\pm 0.3}$ | 705K | $50_{\pm 2}$ |
| SID$_U$ | $4.3_{\pm 1.1}$ | $\mathbf{4.1M}$ | $68_{\pm 2}$ | $5.1_{\pm 1.5}$ | $\mathbf{625K}$ | $\mathbf{39}_{\pm 1}$ | $4.2_{\pm 0.3}$ | 633K | $81_{\pm 3}$ | $3.5_{\pm 0.3}$ | $\mathbf{657K}$ | $63_{\pm 2}$ |
| SID$_C$ | $2.8_{\pm 0.5}$ | $\mathbf{4.1M}$ | $69_{\pm 2}$ | $1.9_{\pm 0.4}$ | $\mathbf{625K}$ | $38_{\pm 3}$ | $\mathbf{3.2}_{\pm 1.1}$ | 633K | $\mathbf{84}_{\pm 2}$ | $3.9_{\pm 0.9}$ | $\mathbf{657K}$ | $\mathbf{66}_{\pm 2}$ |

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

## A    Synthetic datasets

We construct two datasets with the PyTorch Geometric library[3] and the NetworkX library[4], based respectively on Watts-Strogatz, and Erdos-Renyi random graphs. The datasets consist in a list of graphs, where we control the generative factors necessary for synthesizing node feature and adjacency matrices in each graph. This allows us to create labels and latent codes for each graph, similarly to the canonical dataset for disentanglement such as Dsprites[5].

**Node feature factors**    The node feature matrix is computed for both datasets as a random tensor of dimension $N_{\text{nodes}} \times N_{\text{features}}$. The number of node is fixed to 20, while the features are 64. The elements of the feature matrix are sampled from a normal distribution with a given mean and variance, which are two of the generative factors that we control.

**Adjacency matrix factors**    The adjacency matrices are computed via the NetworkX library, where we are able to control the factors responsible for generating the graph.

For Watts-Strogatz graphs we have (1) a factor $k$, which represents the $k$ nearest neighbors in a ring topology, and (2) factor $p$, which is the probability of rewiring each edge. For Erdos-Renyi graphs, we control only one factor $p$, that represents the probability for edge creation.

**Combination**    Watts-Strogatz data has 2 adjacency factors, while Erdos-Renyi has only one, thus, in order to achieve the same number of graphs in both datasets, we fix the variance to 2 on Watts-Strogatz, while we provide 20 values in Erdos-Renyi. The final Watts-Strogatz and Erdos-Renyi datasets have $40 \times 40 \times 20 = 32.000$ graphs. The combinations for the factors: mean, variance, $k$, and $p$ are provided in Table 6. Note that number of nodes is not considered a factor.

Table 6: Graph datasets for disentanglement.

| | | NODE FEATURE | | ADJACENCY MATRIX | | |
| Dataset | Num nodes | Mean | Variance | $k$ | $p$ | Num graphs |
| --- | --- | --- | --- | --- | --- | --- |
| Watts-Strogatz | 20 | 40 | 1 | 20 | 40 | 32K |
| Erdos-Renyi | 20 | 40 | 20 | n/a | 40 | 32K |

## B    Unconditional Disentanglement

The unconditional ELBO (Proposition 1), states that given two latent variables $Z$ and $T$ modeling the log-likelihood of the graph data $\mathcal{G}$ is bounded by the following $\mathcal{L}_{\text{u}}$ ELBO:

$$\mathcal{L}_{\text{u}} = \mathbb{E}_{q(Z|\mathcal{G})q(T|Z)} \log p(\mathcal{G}|Z)p(Z|T) - \mathbb{E}_{q(Z|\mathcal{G})} \text{KL}(q(T|Z)||p(T)) - \mathbb{E}_{q(Z|\mathcal{G})} \log q(Z|\mathcal{G}) \quad (7)$$

*Proof.*  Using Jensen inequality, we have:

$$
\begin{aligned}
\log p(\mathcal{G}) &= \log \int_Z \int_T p(\mathcal{G}, Z, T) \\
&= \log \int_Z \int_T p(\mathcal{G}, Z, T) \frac{q(T|\mathcal{G}, Z)q(Z|\mathcal{G})}{q(T|\mathcal{G}, Z)q(Z|\mathcal{G})} \\
&\geq \int_Z q(Z|\mathcal{G}) \log \int_T p(\mathcal{G}, Z, T) \frac{q(T|\mathcal{G}, Z)}{q(T|\mathcal{G}, Z)q(Z|\mathcal{G})} \\
&= \int_Z q(Z|\mathcal{G}) \log \int_T p(\mathcal{G}, Z, T) \frac{q(T|\mathcal{G}, Z)}{q(T|\mathcal{G}, Z)} - \mathbb{E}_{q(Z|\mathcal{G})} \log q(Z|\mathcal{G})
\end{aligned}
$$

---

[3]https://github.com/pyg-team/pytorch_geometric
[4]https://networkx.org/
[5]https://github.com/deepmind/dsprites-dataset

By applying Jensen inequality again, we obtain:

$$
\begin{aligned}
\log p(\mathcal{G}) &\geq \int_Z q(Z|\mathcal{G}) \int_T q(T|\mathcal{G}, Z) \log \frac{p(\mathcal{G}, Z, T)}{q(T|\mathcal{G}, Z)} - \mathbb{E}_{q(Z|\mathcal{G})} \log q(Z|\mathcal{G}) \\
&= \int_Z q(Z|\mathcal{G}) \int_T q(T|\mathcal{G}, Z) \log \frac{p(\mathcal{G}, Z|T)p(T)}{q(T|\mathcal{G}, Z)} - \mathbb{E}_{q(Z|\mathcal{G})} \log q(Z|\mathcal{G}) \\
&= \mathbb{E}_{q(Z|\mathcal{G})} \mathbb{E}_{q(T|\mathcal{G}, Z)} \log p(\mathcal{G}, Z|T) - \mathbb{E}_{q(Z|\mathcal{G})} \int_T q(T|\mathcal{G}, Z) \log \frac{q(T|\mathcal{G}, Z)}{p(T)} - \mathbb{E}_{q(Z|\mathcal{G})} \log q(Z|\mathcal{G}) \\
&= \mathbb{E}_{q(Z|\mathcal{G})} \mathbb{E}_{q(T|\mathcal{G}, Z)} \log p(\mathcal{G}, Z|T) - \mathbb{E}_{q(Z|\mathcal{G})} \mathrm{KL}(q(T|\mathcal{G}, Z)||p(T)) - \mathbb{E}_{q(Z|\mathcal{G})} \log q(Z|\mathcal{G})
\end{aligned}
$$

The results follows assuming $p(\mathcal{G}|Z, T) = p(\mathcal{G}|Z)$, and noting that for computing $T$, we only use the adjacency information $\mathcal{A}$ from $\mathcal{G}$.

$\square$

## C   Conditional Disentanglement

Using Jensen's inequality, we note that for any fixed $\hat{Z}$

$$
\begin{aligned}
\log p(\mathcal{G}, \hat{Z}) &= \log \int_Z \int_T p(\mathcal{G}, Z, T, \hat{Z}) \\
&= \log \int_Z \int_T p(\mathcal{G}, Z, T, \hat{Z}) \frac{q(Z, T|\mathcal{G})}{q(Z, T|\mathcal{G})} \\
&= \log \int_Z \int_T p(\mathcal{G}, Z, T, \hat{Z}) \frac{q(T|\mathcal{G}, Z)q(Z|\mathcal{G})}{q(T|\mathcal{G}, Z)q(Z|\mathcal{G})} \\
&\geq \int_Z q(Z|\mathcal{G}) \log \int_T p(\mathcal{G}, Z, T, \hat{Z}) \frac{q(T|\mathcal{G}, Z)}{q(T|\mathcal{G}, Z)q(Z|\mathcal{G})} \\
&= \int_Z q(Z|\mathcal{G}) \log \int_T p(\mathcal{G}, Z, T, \hat{Z}) \frac{q(T|\mathcal{G}, Z)}{q(T|\mathcal{G}, Z)} - \mathbb{E}_{q(Z|\mathcal{G})} \log q(Z|\mathcal{G})
\end{aligned}
$$

Since we use only $\mathcal{A}$ from $\mathcal{G}$ for computing $T$, we can write $q(T|\mathcal{G}, Z) = q(T|Z, \mathcal{A})$, and applying Jensen's inequality again, we get

$$
\begin{aligned}
\log p(\mathcal{G}, \hat{Z}) &\geq \int_Z q(Z|\mathcal{G}) \log \int_T p(\mathcal{G}, Z, T, \hat{Z}) \frac{q(T|Z, \mathcal{A})}{q(T|Z, \mathcal{A})} - \mathbb{E}_{q(Z|\mathcal{G})} \log q(Z|\mathcal{G}) \\
&\geq \int_Z q(Z|\mathcal{G}) \int_T q(T|Z, \mathcal{A}) \log \frac{p(\mathcal{G}, Z, T, \hat{Z})}{q(T|Z, \mathcal{A})} - \mathbb{E}_{q(Z|\mathcal{G})} \log q(Z|\mathcal{G}) \\
&= \mathbb{E}_{q(Z|\mathcal{G})} \mathbb{E}_{q(T|Z, \mathcal{A})} \log \frac{p(\mathcal{G}, Z, T, \hat{Z})}{q(T|Z, \mathcal{A})} - \mathbb{E}_{q(Z|\mathcal{G})} \log q(Z|\mathcal{G}) \\
&= \mathbb{E}_{q(Z|\mathcal{G})} \mathbb{E}_{q(T|Z, \mathcal{A})} \log \frac{p(Z, T)p(\hat{Z}|Z, T)p(\mathcal{G}|\hat{Z}, T, Z)}{q(T|Z, \mathcal{A})} - \mathbb{E}_{q(Z|\mathcal{G})} \log q(Z|\mathcal{G}) \\
&= \mathbb{E}_{q(Z|\mathcal{G})} \mathbb{E}_{q(T|Z, \mathcal{A})} \log \left( p(\hat{Z}|Z, T)p(\mathcal{G}|\hat{Z}, T, Z) \right) - \mathbb{E}_{q(Z|\mathcal{G})} \mathrm{KL}(q(T|Z, \mathcal{A})||p(T, Z)) \\
&\quad - \mathbb{E}_{q(Z|\mathcal{G})} \log q(Z|\mathcal{G}) .
\end{aligned}
$$

Assuming $\mathcal{G}$ is independent of $Z$ and $T$ given $\hat{Z}$, we immediately get

$$
\begin{aligned}
\log p(\mathcal{G}, \hat{Z}) &\geq \mathbb{E}_{q(Z|\mathcal{G})} \mathbb{E}_{q(T|Z, \mathcal{A})} \log \left( p(\hat{Z}|Z, T)p(\mathcal{G}|\hat{Z}) \right) - \mathbb{E}_{q(Z|\mathcal{G})} \mathrm{KL}(q(T|Z, \mathcal{A})||p(T, Z)) \\
&\quad - \mathbb{E}_{q(Z|\mathcal{G})} \log q(Z|\mathcal{G}) .
\end{aligned}
$$

$\square$

## D   Scalability experiment

We perform a graph classification task on large scale datasets, and report their dimensions in Table 7. In Table 8 we report the results for our $SID_C$ model on social network datasets including IMDB binary IMDB-multi, COLLAB, as well as macro molecules such as PROTEINS and MUTAG. The baselines for this experiment include FactorGCN [66] and DGCL [31]. For our model we set group-size to 81, hessian penalty to 40 and commutative penalty to 5. We follow the classification setup from Yang et al. [66], with a ten-fold cross-validation procedure, and report accuracy and standard deviation. The results show that both $SID_U$ and $SID_C$ are able to improve the performance of the baselines. In particular, $SID_C$ achieves the highest accuracy on all the datasets.

Table 7: Dimensions of datasets

|  | IMDB-B | IMDB-M | COLLAB | PROTEINS | MUTAG |
|---|---|---|---|---|---|
| Graphs | 1000 | 500 | 5000 | 1113 | 188 |
| Classes | 2 | 3 | 3 | 2 | 2 |
| Avg. Nodes | 19.77 | 13.00 | 74.49 | 39.06 | 17.93 |
| Avg. Edges | 96.53 | 65.94 | 2457.78 | 72.82 | 19.79 |

Table 8: Scalability

|  | IMDB-B | IMDB-M | COLLAB | PROTEINS | MUTAG |
|---|---|---|---|---|---|
| FactorGCN | 75.3 ± 2.7 | - | 81.2 ± 1.4 | - | 89.9 ± 6.5 |
| DGCL | 75.9 ± 0.7 | 51.9 ± 0.4 | 81.2 ± 0.3 | 76.4 ± 0.5 | 92.1 ± 0.8 |
| $SID_U$ | 76.1 ± 0.2 | 51.2 ± 0.9 | 81.9 ± 0.8 | 76.5 ± 1.5 | 91.4 ± 1.2 |
| $SID_C$ | **76.5 ± 0.3** | **52.5 ± 0.2** | **82.5±0.2** | **76.9 ± 0.1** | **92.5 ± 0.5** |

## E   Molecular experiment

In order to have a fair comparison with Flow models, such as GraphAF [55] and GraphDF [37], we incorporate our parameterization into the training process of a Flow model. Both GraphAF and GraphDF are designed using relational GCN [53] (RGCN) as building block to compute an embedding of a graph using a layer R-GCN: $H_i^L = $ R-GCN$(G_i)$, $\tilde{h}_i = $ sum$(H_i^L)$ where sum denotes the sum-pooling operation, and $H_{i,j}^L \in \mathbb{R}^k$ denotes the embedding of the $j$-th node in the embeddings $H_i^L$.

We incorporate some parts of our models after the RGCN encoding, as follows. From the embedding $H$, we compute the lie algebra coordinates, with an encoder $q(T|H)$, following our $E_{group}$ network from Section 4. We then apply the lie algebra exponential mapping to reconstruct the embedding $H$ as $p(H|T)$, following the $D_{group}$ network from Section 4. Furthermore, we incorporate the hessian penalty as a regularization. In all the three tasks, we follow the experimental setup from GraphAF, we set the model with group-size of 81, and a hessian penalty of 40.

We first perform the random graph generation task, which involves quantifying 4 metrics in percentage, including: valid molecules with resampling (Val), valid molecules without resampling (Res), unique molecules (Uni), and novel molecules not appearing in the training data (Nov). In Table 9 we report results for GCPN, GraphAF, GraphDF and our models. The results indicate that with the inclusion of a lie algebra reparameterization and the hessian penalty, the performance on random generation is enhanced, and the Flow + $SID_C$ model achieves the best performance on all metrics a part from uniqueness, where it achieves comparable results.

Secondly, we report the results for the property optimization task in Table 10, which show the scores from the top 3 generated molecules for two selected properties (penalized logp and QED). The baselines are the same ones as in the random generation task. We observe that our Flow + $SID_C$ model is the top performing method on penalized logP for the top 3 scores, and it achieves the top results, together with the other methods, for QED.

Table 9: Random graph generation.

| Model | ZINC | | | | QM9 | | | | MOSES | | | |
|---|---|---|---|---|---|---|---|---|---|---|---|---|
| | Val | Res | Uni | Nov | Val | Res | Uni | Nov | Val | Res | Uni | Nov |
| GCPN | **100** | 20 | **99.97** | **100** | n/a | n/a | n/a | n/a | n/a | n/a | n/a | n/a |
| GraphAF | **100** | 68 | 99.1 | **100** | **100** | 67 | 94.15 | 88.83 | **100** | 71 | **99.99** | **100** |
| GraphDF | **100** | 89.03 | 99.16 | **100** | **100** | 82.67 | **97.62** | 98.1 | **100** | 87.58 | 99.55 | **100** |
| $SID_U$ | **100** | 78.21 | 99.12 | **100** | **100** | 75.43 | 95.22 | 90.33 | **100** | 81.62 | 99.62 | **100** |
| $SID_C$ | **100** | 86.02 | 99.14 | **100** | **100** | 82.92 | 96.43 | 92.28 | **100** | 84.29 | 99.79 | **100** |
| Flow + $SID_C$ | **100** | **90.12** | 99.53 | **100** | **100** | **83.45** | 97.46 | **98.62** | **100** | **88.42** | 99.79 | **100** |

Table 10: Property optimization performance.

| | Penalized logP | | | QED | | |
|---|---|---|---|---|---|---|
| | 1st | 2nd | 3rd | 1st | 2nd | 3rd |
| ZINC | 4.52 | 4.3 | 4.23 | 0.948 | 0.948 | 0.948 |
| GCPN | 7.98 | 7.85 | 7.80 | **0.948** | 0.947 | 0.946 |
| GraphAF | 12.23 | 11.29 | 11.05 | **0.948** | **0.948** | 0.947 |
| GraphDF | 13.7 | 13.18 | 13.17 | **0.948** | **0.948** | **0.948** |
| $SID_U$ | 12.56 | 12.46 | 12.03 | 0.947 | **0.948** | 0.947 |
| $SID_C$ | 12.89 | 12.82 | 12.27 | **0.948** | **0.948** | **0.948** |
| Flow + $SID_C$ | **13.97** | **13.35** | **13.39** | **0.948** | **0.948** | **0.948** |

Finally, we show in Table 11 the results for the constrained optimization task. We report only GraphDF as baseline, which is the top performing model among the baselines. The evaluation involves reporting the mean and standard deviation of metrics including: the largest property improvement (Imp), and similarities (Sim) between them and their corresponding input molecules, as well as the success rate (Suc).

We observe that by combining the disentangled principles developed in our models into a Flow-based model, we are able to further improve the results, and achieve the top performance.

Table 11: Constrained optimization performance on 800 molecules used in GraphAF.

| $\delta$ | Flow + $SID_C$ | | | GraphDF | | | $SID_C$ | | |
|---|---|---|---|---|---|---|---|---|---|
| | Imp | Sim | Suc | Imp | Sim | Suc | Imp | Sim | Suc |
| 0.0 | **14.28**$_{\pm 5.24}$ | 0.29$_{\pm 0.10}$ | 100 | 14.15$_{\pm 6.86}$ | 0.29$_{\pm 0.13}$ | 100 | 13.26$_{\pm 3.24}$ | 0.28$_{\pm 0.21}$ | 100 |
| 0.2 | **12.89**$_{\pm 4.33}$ | 0.34$_{\pm 0.09}$ | 99.93 | 12.77$_{\pm 6.59}$ | 0.32$_{\pm 0.11}$ | 100 | 12.35$_{\pm 5.62}$ | 0.32$_{\pm 0.13}$ | 100 |
| 0.4 | **9.32**$_{\pm 5.45}$ | 0.50$_{\pm 0.03}$ | 99.3 | 9.19$_{\pm 6.43}$ | 0.48$_{\pm 0.08}$ | 99.6 | 8.25$_{\pm 4.93}$ | 0.50$_{\pm 0.18}$ | 98.4 |
| 0.6 | **4.98**$_{\pm 6.21}$ | 0.67$_{\pm 0.04}$ | 97.3 | 4.51$_{\pm 5.80}$ | 0.65$_{\pm 0.05}$ | 92.1 | 4.67$_{\pm 5.39}$ | 0.66$_{\pm 0.03}$ | 93.2 |