# OpenReview forum: "Symmetry-induced Disentanglement on Graphs"
_NeurIPS.cc/2022/Conference — NeurIPS 2022 Accept_

### Official Review · Reviewer_FmQN · 2022-07-09

**Rating:** 8
**Confidence:** 5
**Soundness:** 4 excellent
**Presentation:** 3 good
**Contribution:** 3 good

**Summary:**

This paper brings group-based disentanglement learning to graph data domain. Two disentanglement cases, unconditional and conditional, are defined under this framework in this paper. The proposed architecture encodes group data into group latent representations with Lie algebra and exponential map, and leverages techniques like Hessian penalty to enforce disentanglement. Experiments on various data have shown the model’s effectiveness in disentanglement on graph data.

**Questions:**

Please see weaknesses.

**Limitations:**

Yes

**Strengths And Weaknesses:**

Pros:
1. The idea of group-based disentanglement is relatively new, and it is interesting to see it being brought to the graph data domain.
2. The proposed architecture is quite simple and the shown experimental results validate its effectiveness.

Cons:
1. In Fig. 1 right, why is the addition from Z is used in the conditional case? Does it mean Z will contain extra information that is not encoded in the latent T?
2. Is it possible to qualitatively show what information has been disentangled in the graph data?
3. The section 2.2 GNN architecture part seems a little redundant to me as Sec. 3.2 and 3.3 will introduce the architecture in detail.

---

> ### Author Response · Authors · 2022-08-02
> **Discussion and experiments**
>
> ##### Information contained by $Z$ and $T$ in conditional disentanglement.
> Thanks for raising this important question. Yes, in the conditional disentanglement setting, $Z$ contains information pertaining to both the disentangled and entangled factors, while $T$ contains only information pertaining to the disentangled part. Specifically, the group encoder $E_{group}$ takes $Z$ as input and obtains vector $T$ that contains the $k$ coordinates of a Lie algebra. Thus $T$ contains the disentangled factors. However, it misses out on the information contained in the $Z$ that could not be disentangled. The addition from $Z$ is therefore needed to ensure that we recover this information  as well, so that we can subsequently reconstruct the graph with the decoder $D_{graph}$ that uses both the uncoupled and the entangled parts.
>
> ##### Visualizations/Qualitative evidence for disentanglement.
> Thanks for this excellent suggestion! We have now added the results of an experiment - as Appendix $F$ in the Supplementary - to provide qualitative evidence for disentanglement.
>
> Specifically, in this setup, we trained the proposed conditional model (CON) on  Watts-Strogatz (WS) random graphs. These graphs have 3 generative factors:  the number of nodes $n$,  connectivity (i.e., the probability of rewiring each edge) $p$, and the neighborhood size $k$ (pertaining to $k$ nearest neighbors in a ring topology). We also sampled a random node attribute value (or color) $v_i$ between $[0, 1]$ independently for each node $i$. Clearly, while $n$, $p$, and $\{v_i\}$ are independent, $k$ is entangled with $p$.  We then followed (Stoehr et al. 2019)[1] to project latent variables in pairs to see how the factors vary with each other. We're interested in disentangling $n$, $p$, and $\{v_i\}$ using the proposed conditional model CON.
>
> In Figure 5 in the Supplementary, we report the traversal of the CON trained using 3 latent variables, and we observe the following.
>
> - The variable $z_0$ controls the node attribute value which is indicated by the color in the right column. For example, in the top row on the right, we see the color changes from blue to white along the horizontal axis.
> - The variable $z_1$ controls connectivity, for example on top row, following the $y$ axis, we see that the edges decrease, and in second row, along the $x$ axis, edges increase.
> - The variable $z_2$ controls the number of nodes, and we see that in the middle row and the last row the number of nodes decreases along the vertical axis.
> Clearly, the CON model has effectively learnt effective disentangled representations.
>
> ##### Removing redundancy by merging sections.
> Thanks for your comments regarding redundancy of these parts. As you suggested, we will remove/shorten section 2.2, or merge it with sections 3.2 and 3.3 that detail the description of GNN architectures to eliminate any repetitions.
>
> Thanks, again, for your very constructive feedback. We hope your concerns and questions have been sufficiently addressed, and if so, the same translates into raised scores.
>
> References
> [1] Stoehr et al. 2019 - Disentangling interpretable generative parameters of random and real-world graphs

---

> > ### Comment · Reviewer_FmQN · 2022-08-07
> > **Thank Authors for the Response**
> >
> > Thank you for the response. My concerns are addressed clearly. I will raise my rating for this paper.

---

> > > ### Author Response · Authors · 2022-08-09
> > > **Thanks for your response**
> > >
> > > Many thanks for your response! We’re glad that your concerns have been addressed, and are grateful for your strong support.

---

### Official Review · Reviewer_Em3T · 2022-07-11

**Rating:** 8
**Confidence:** 3
**Soundness:** 4 excellent
**Presentation:** 4 excellent
**Contribution:** 4 excellent

**Summary:**

The authors first propose a formalization of graph disentanglement in terms of graph embeddings building on existing definitions. They then propose a graph+group VAE architecture incorporating the graph disentanglement definition. This autoencoder first uses a graph encoder to map the original graph features to k latent features, which are then independently passed to k GCNs and aggregated by the group encoder to obtain the group coordinates. They are then decoded by the group decoder, combining the basis elements and passing them to the graph decoder. The authors then demonstrate significant improvements in several metrics, particularly disentanglement metrics.

**Questions:**

How universal is the proposed definition - are there classes of graphs for which it may not be ideal?

What are some practical implications of this kind of disentanglement on graphs? In what contexts might this be useful?

**Limitations:**

Aside from what is discussed above, I think the authors reasonably discussed their limitations.

**Strengths And Weaknesses:**

The paper in general is clear, well-written, well-motivated, and sufficiently rigorous. Although I am not especially familiar with previous work on graph disentanglement in particular, having reviewed the related works, the proposed method seems novel in the context of graphs. The translation of their proposed definition to their GVAE seems particularly clever and does not seem to obviously follow from the definition alone: it captures the key features of their definition while still being highly general and customizable. The experiments are fairly convincing as well, with some impressive disentanglement performance on the datasets analyzed (though more disentanglement datasets would have been helpful).

My primary (albeit limited) concern with this paper’s definition is that it appears to more-or-less directly translate the concept of disentanglement in terms of underlying generative latent variables to graphs, embeddings and all, while seemingly largely disregarding graph-specific elements. However, graphs are potentially meaningfully different from the unstructured spaces for which disentanglement has previously been formalized. For example, graphs have often been used to express latent variable models, of which VAEs are a specific kind - though latent variable graphs are only a specific kind, it is not obvious that this framework would be the best way to characterize disentanglement in contexts where the graphs themselves represent generative processes. This is not to say that the proposed definition is not useful in this context: generative processes can themselves be the product of an underlying generative process. However, it is likely worth exploring and discussing the kinds of graphs where this definition may not cleanly apply.

In addition, it would also be helpful to provide some additional justification for the metrics used. While they are standard disentanglement metrics, it does not appear that many prior works specifically on graphs aside from a 2019 workshop paper use them (including the ones cited). In particular, a workshop paper that may be worth mentioning is “Disentangling Interpretable Generative Parameters of Random and Real-World Graphs” (Stoehr et al 2019) which much less formally proposes evaluating graph disentanglement in terms of equivariance between the underlying generative and corresponding encoding maps. Finally, some visualizations of the kinds of variations learned by disentangled models would have been helpful.

---

> ### Author Response · Authors · 2022-08-02
> **Discussion**
>
> ##### Justification of metrics
>
> Thank you for raising this up. Indeed, previous work doesn't quite evaluate quantitatively disentanglement in the representations, except for FactorGCN, that proposed a reasonable metric.
>
> In contrast, we considered a wide array of experiments and evaluation metrics that arise quite naturally in certain practical applications that the community cares about. For example, QED and Penalized logP are aligned with a molecular optimization perspective, where starting from latent code of a possibly sub-optimal molecule, we seek to navigate to molecules that exhibit better properties. Disentanglement becomes quite important in such scenarios since, e.g., we might want to suppress toxicity but retain other desirable properties. Similarly, we showed that metrics from vision based datasets, like d-sprite, can be meaningful in graph settings, and to help understand the degree of disentanglement in GNNs.
>
> Based on your suggestion, we would emphasize the relevance of different metrics considered in this work.
>
> ##### Visualizations of variations
> Thank you for pointing us to (Stoehr et al 2019)[1] and for your excellent suggestion about visualizations! We will be sure to appropriately position the impressive contributions by (Stoehr et al.) in the final version.
>
> We have now added the results of an experiment - as Appendix $F$ in the Supplementary - to provide qualitative evidence for disentanglement.
>
> Specifically, in this setup, we trained the proposed conditional model (CON) on  Watts-Strogatz (WS) random graphs. These graphs have 3 generative factors:  the number of nodes $n$,  connectivity (i.e., the probability of rewiring each edge) $p$, and the neighborhood size $k$ (pertaining to $k$ nearest neighbors in a ring topology). We also sampled a random node attribute value (or color) $v_i$ between $[0, 1]$ independently for each node $i$. Clearly, while $n$, $p$, and $\{v_i\}$ are independent, $k$ is entangled with $p$.  We then followed (Stoehr et al. 2019)[1] to project latent variables in pairs to see how the factors vary with each other. We're interested in disentangling $n$, $p$, and $\{v_i\}$ using the proposed conditional model CON.
>
> In Figure 5 in the Supplementary, we report the traversal of the CON trained using 3 latent variables, and we observe the following.
>
> - The variable $z_0$ controls the node attribute value which is indicated by the color in the right column. For example, in the top row on the right, we see the color changes from blue to white along the horizontal axis.
> - The variable $z_1$ controls connectivity, for example on top row, following the $y$ axis, we see that the edges decrease, and in second row, along the $x$ axis, edges increase.
> - The variable $z_2$ controls the number of nodes, and we see that in the middle row and the last row the number of nodes decreases along the vertical axis.
>
> Certainly, for our synthetic setting, the CON model learned effective disentangled representations.
>
>
> ##### On the universality of proposed definition
> Thank you for your incisive comments and questions on this subject. Indeed, one of the main objectives of this work has been to go beyond the existing notion of unconditional disentanglement, via a conditional definition that allows for some entangling in the latent space.
>
> That said, as you elegantly articulated, this notion of conditional disentanglement might not suffice for certain settings and applications. For example, causal graphs afford the ability to reason about questions such as ``what if someone had done A instead of B". However, learning the structure of the causal graph from observational data alone is generally not possible. The conditional notion of disentanglement proposed here does not accommodate interventional data, so expecting it to able to disentangle factors on a graph that may have complex causal dependencies would not be advisable. However, the present work does open up avenues for possible extensions in such scenarios.
>
> We will be sure to add a discussion on this.
>
> ##### Practical implications
> We believe this work has significant practical implications. For example, the proposed model performed extremely well on the problem of property optimization. This is remarkable since our model is not optimized for molecule generation.  Thus disentangled representations could form a vital role in applications such as drug discovery, where  encouraging the separation of generative factors into different latent
> codes has the potential to improving the quality of generated molecules. Similarly, it opens up new possibilities for other hard problems involving multi-objective optimization such as material synthesis. \\
>
>  Moreover, disentanglement becomes important in the context of interpretable models since human often require explainable factors for several critical applications. We will discuss such implications in the final version.

---

> > ### Comment · Reviewer_Em3T · 2022-08-07
> > **Thanks for the response!**
> >
> > Thank you for this additional detail and discussion! I appreciate the additional discussion and examples and believe they reinforce the paper's results and presentation so I have raised my score.

---

> > > ### Author Response · Authors · 2022-08-09
> > > **Thanks**
> > >
> > > Thank you so much! We’re grateful for your response, and for your strong support for the proposed work.

---

### Official Review · Reviewer_DqMr · 2022-07-13

**Rating:** 6
**Confidence:** 3
**Soundness:** 4 excellent
**Presentation:** 3 good
**Contribution:** 3 good

**Summary:**

The paper proposes two (conditional and unconditional) symmetry-based and Lie-algebras-based disentanglement on Graph Neural Networks within the framework of Variational Autoencoder. As is defined in symmetry-based disentanglement, a symmetry group, which can be decomposed into subgroups, can help disentangle a vector representation affected by it if specific demands are satisfied. Therefore, the authors introduce a Lie group as the symmetry group to graph disentanglement. Besides, they add commutative penalty and hessian penalty to enforce further disentanglement. In the experiment part, they evaluate systematic disentanglement on multiple metrics based on synthetic datasets and model performance based on real datasets.

**Questions:**

1.	There are two minor errors in Conclusion. In line 319, “and” should be “an”. In line 320, “real-word” should be “real-world”.

**Strengths And Weaknesses:**

Strengths:
1.	The disentanglement on graph is an important topic and the paper provides a new insight from symmetry-based one and introduces the Lie group to it.
2.	The consideration of commutative penalty, hessian penalty is comprehensive.

Weaknesses:
1.	As is written in Conclusion, this paper doesn’t present the disentangled factors on real-world datasets. But both [1] and [2], which are mentioned in Related Work, have disentanglement evaluations based on feature correlation analysis. Therefore, it is possible to evaluate disentanglement on real-world datasets, although it is still open which one is the most convincing.
[1] Jianxin Ma, Peng Cui, Kun Kuang, Xin Wang, and Wenwu Zhu. Disentangled graph convolutional networks. In International conference on machine learning, pages 4212–4221. PMLR, 2019.
[2] Yiding Yang, Zunlei Feng, Mingli Song, and Xinchao Wang. Factorizable graph convolutional networks. Advances in Neural Information Processing Systems, 33:20286–20296, 2020.

---

> ### Author Response · Authors · 2022-08-02
> **Correlation analysis**
>
> Thanks for your excellent suggestions.
> As you suggested, we added a feature correlation analysis in appendix E, following the dataset from Yang et al [1], where each graph is composed by 6 factor graphs. The correlation matrix in Figure 4 aims to show whether the features capture mutually exclusive information, which is achieved when the there is a block-wise correlation pattern. Both our UNC and COND models achieve a block-wise correlation pattern, however the conditional version has also some other highlighted regions, capturing the entangled factors. In terms of setup, our models use a group-size of 32 (to match with hidden features size of the other methods), an hessian penalty of 40 and a commutative penalty of 5.
>
>
> References
> [1] Yang et al NeurIPS 2020 - Factorizable graph convolutional networks

---

> > ### Author Response · Authors · 2022-08-09
> > **Thanks for the suggestions**
> >
> > Thanks, again, for your constructive feedback. Based on your suggestions, we have performed disentangled evaluations pertaining to feature correlation analysis using the dataset and setting of Yang et al. (please see Appendix E and Fig. 4). Please also see the new scalability experiments with several large-scale real datasets (Appendix D and Table 8)  that reinforce the efficacy of the proposed method. We’ve also fixed the minor edits you suggested. We would be grateful if you could kindly acknowledge the same, and consider raising your rating for the paper. Many thanks!

---

### Official Review · Reviewer_3Dsd · 2022-07-15

**Rating:** 5
**Confidence:** 3
**Soundness:** 2 fair
**Presentation:** 2 fair
**Contribution:** 2 fair

**Summary:**

This paper emphasizes learning disentangled representations from graphs. In contrast to prior works, the authors formulate new definitions of unconditional/conditional symmetry-based graph disentanglements and developed a Lie group-based framework for learning with Graph VAE. The authors evaluate their methods on disentanglement metrics constructed on synthetic datasets and downstream tasks, including few-shot classiﬁcation, compression and molecular graph generation.

**Questions:**

**[Experimental setup]**

- The derivation in section 3.2 use a single 2-layer GCN, does this architecture is used across all the experiments in section 5.3? I didn't find a clear description in the section. 5 nor the appendix.
- The setting value of the communitative penalty is not mentioned.
- What are the comparisons on the similarity and success rates (in line 307)? Are all the methods follow the same setting of distance threshold  $\delta$ ? How to control that threshold in JT-VAE?

**[Interpretations MIA on Table.2]** Its turns out MIA is a strong baseline that achieves the best MSE performance across all the datasets. Any interpretations on this?

**[Scaling Up]** How would the proposed method perform on large graph task (macromolecules, community graphs)?

**Limitations:**

The authors adequately addressed the limitations (disentanglements can be measured only on synthetic datasets) of their work in section.6

**Strengths And Weaknesses:**



- ***Originality***: Are the tasks or methods new? Is the work a novel combination of well-known techniques? (This can be valuable!) Is it clear how this work differs from previous contributions? Is related work adequately cited.

  Learning disentangled representaitons on graphs is a

- ***Quality***: The work is complete, my concerns on the theoretical analysis and experimental results are put at the *clarity* and **Questions**.

- ***Clarity***: The submission is clearly written and well organized. The experimental descriptions are not clear to me. My suggestions are as follows:

  **[Adjacency in GNN architechture]** In lines 88-99, the notation of adjacency should be introduced at the beginning, otherwise how to conduct the graph convolution proceducre? The current formuation sees to be have space for improving readability.

  **[Spelling]** (2 c) -> (2 classes) in line 252; in line 319 "Since this is and open problem in representation learning" is not a sentence, need to be rephrased.

- ***Significance***: With the code provided in the supplementary, others researchers or practitioners can use the ideas or build on them. The constructed dataset should also be released (lines 217-219). Besides demonstrating the effectiveness of the proposed method, I have a question about the weakness of comparisons in the molecular generation and property optimization tasks. It seems that all the ways perform equally well/bad in these tasks, and many scores are precisely the same. Given the usages of different models and methods, it is a bit unclear to validate the improvements induced by disentangling the learned representations on these tasks.

---

> ### Author Response · Authors · 2022-08-02
> **Added experiments, clarified setup, provided interpretation**
>
> Thanks for your detailed comments to improve the paper.
>
> Updates
> - GNN Adjacency: we updated the "GNN architecture" paragraph following your comment.
> - Significance: As you kindly pointed out our method is not achieving the best result on molecular tasks. We note that our model already can get better results from other models, such as JT-VAE, GCPN, graphAF[1] for both random generation (table 3) and property optimization (table 4). The CON model also achieves better performance than graphAF on constrained optimization (table 5).  We performed an additional experiment, on molecular datasets, reported in Appendix G, where we show that disentanglement can improve the performance of a flow model to further clarify the significance of our method. Our model setup involves incorporating our lie-algebra parameterization into the training process of a Flow model. More specifically, insert our component after the flow model has encoded the graph data into an embedding. In fact, both GraphAF[1] and GraphDF[2] are designed to use a relational GCN [3] as building block to compute an embedding of a graph. The embedding is given as: $R-GCN: H_i^{L} = R-GCN(G_i)$  $\tilde{h_i} = sum(H_i^L)$  where sum denotes the sum-pooling operation, and $H_{i,j}^L \in \mathbb{R}^k$ denotes the embedding of the $j$-th node in the embeddings $H_i^L$. From the embedding $H$, we compute the lie algebra coordinates, with an encoder $q(T|H)$, following our $\text{E}_\text{group}$ network from Section 3. We then apply the lie algebra exponential mapping  to reconstruct the embedding $H$ as $p(H|T)$, following the $\text{D}_\text{group}$ network from Section 3. Furthermore, we incorporate the hessian penalty as a regularization. We follow the same experimental setup as in our previous molecular generation experiment. The experiments in Appendix G (tables 9 - 10 -11) show that incorporating this mechanism in a flow model (GraphAF [1]) can lead to performance improvements.
> - Experimental setup: Thanks for finding these issues. We updated the paper with the following info: 1) Layers: We stack 3 of the defined layers for the experiments, where each layer takes the graph features outputs from the previous one as input - see Sec 3.4. 2) Commutativity penalty: is set to 5 on all experiments. 3) Molecular experiments setup: We follow the setups from previous work, namely GraphAF [1], and we take the results of all baselines from their papers. In terms of Success rate and similarity - our model achieves comparable results on all the explored $\delta$ values.
> - Scaling up: Thanks for raising this important point. We added a new graph classification task on large scale datasets - reported in Appendix D. In Table 7 of (appendix D) we report the results for our COND model on social network datasets, including IMDB binary IMDB-multi, COLLAB, as well as macro molecules such as PROTEINS and MUTAG. We compare with FactorGCN (Yang et al. 2020)[7] and DGCL (Li et al. 2021)[6]. We use group-size 81, hessian penalty 40 and commutative penalty of 5. This experiment shows that our model is able to outperform other baselines for graph disentanglement.
> - Interpretation on MIA: Thanks for raising this issue. We believe that the problem is that in disentangled VAEs there can be a trade-off between reconstruction and disentanglement. This phenomenon has been shown in previous work on disentanglement (betaVAEs)[5]. We notice that the MIA model is an Autoencoder - not a VAE - and it does not learn disentangled representations. We believe that this is the main reason for the high MSE in our model. Furthermore, we observe in Table 2 that in the Mutagenicity dataset we are able to get the best performance with the CON model, and in the COLOR-3 dataset CON achieves comparable MSE, while on NCI1 both UNC and CON achieve similar values compared to MIA.
>
>
> References
> [1] Luo et al- ICLR 2020  GraphAF: a Flow-based Autoregressive Model for Molecular Graph Generation
> [2] Shi et al - ICML 2021 GraphDF: A discrete flow model for molecular graph generation
> [3] Schlichtkrull et al 2018 - Modeling relational data with graph convolutional networks
> [4] Ge et al 2021 - Graph Autoencoder for Graph Compression and Representation Learning
> [5] Higgins et al ICLR 2017 - beta-vae: Learning basic visual concepts with a constrained variational framework
> [6] Li et al NeurIPS 2021 - Disentangled Contrastive Learning on Graphs
> [7] Yang et al NeurIPS 2020 - Factorizable graph convolutional networks

---

> > ### Comment · Reviewer_3Dsd · 2022-08-08
> > **Thanks for the response.**
> >
> > Thanks for your response. My concerns are partially addressed. I remain my score as leaning for acceptance.

---

> > > ### Author Response · Authors · 2022-08-09
> > > **Thanks for the discussion**
> > >
> > > Thank you – we appreciate your engagement in the discussion. Could you please be specific about which of your concerns is still not addressed adequately? Thanks, also, for your questions and suggestions (in particular, the scalability experiment) –  that have helped reinforced the strengths of the paper – please take a look at the new evidence – so, we would appreciate if the same is translated into a revised score.

---

### Meta-Review · Area_Chair_bDJt · 2022-08-23

**Recommendation:** Accept
**Confidence:** Less certain

**Metareview:**

The paper presents an algorithm for learning disentangled representations for graph data, in the graph VAE framework. The reviewers appreciate the cleanly defined problem formulation with background knowledge (sec 2), which leads to a practical training objective and regularization (sec 3). Experimental results show that better disentanglement metrics are obtained with the proposed method compared to alternative ones, and better few short classification performance is achieved indicating important structures being learned to provide better generalization. The authors are encouraged to discuss the computational complexity aspects of the method, and provide visualization to help readers interpret group operations on graph and symmetry.

**Award:**

No

---

### Decision · Program_Chairs · 2022-09-14

Accept